# Experimental Characterization of Water Condensation Processes on Self-Assembled Monolayers Using a Quartz Crystal Microbalance with Energy Dissipation Monitoring

Subin Song [1], Glenn Villena Latag [1], Evan Angelo Quimada Mondarte [2],*, Ryongsok Chang [1] and Tomohiro Hayashi [1],*

[1] Department of Material Science and Engineering, School of Materials and Chemical Technology, Tokyo Institute of Technology, 4259 Nagatsuta-Cho Midori-Ku, Kanagawa, Yokohama 226-8502, Japan

[2] School of Materials Science and Engineering, Nanyang Technological University, Singapore 639798, Singapore

* Correspondence: evanangelo.mondarte@ntu.edu.sg (E.A.Q.M.); tomo@mac.titech.ac.jp (T.H.);
Tel.: +65-9081-8490 (E.A.Q.M.); +81-45-924-5400 (T.H.)

**Abstract:** Water condensation on solid surfaces is a universal phenomenon that plays an essential role in many interfacial phenomena, such as friction, corrosion, adsorption, etc. Thus far, the initial states of water condensation on surfaces with varying chemical properties have yet to be fully explained at the nanoscale. In this study, we performed a real-time characterization of water condensation on self-assembled monolayers (SAMs) with different functional groups using quartz crystal microbalance with energy dissipation monitoring (QCM-D). We found that the kinetics of water condensatison is critically dependent on the head group chemistries. We discovered that the condensed water's viscoelasticity cannot be predicted from macroscopic water contact angles, but they were shown to be consistent with the predictions of molecular simulations instead. In addition, we also found a highly viscous interfacial water layer on hydrophilic protein-resistant SAMs. In contrast, the interfacial water layer/droplet on either hydrophilic protein-adsorbing or hydrophobic SAMs exhibited lower viscosity. Combining our and previous findings, we discuss the influence of interfacial hydration on the viscoelasticity of condensed water.

**Keywords:** condensation; viscoelasticity; interfacial water; interfacial hydration; functional surface; QCM-D

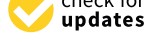



## 1. Introduction

Water condensation is the phase-change process where water vapor turns into liquid water. Recent theoretical and experimental studies have revealed that nanoscale water layers are present even on seemingly dry surfaces [1,2]. The trace amounts of water play an essential role in various interfacial phenomena, such as adsorption [3], corrosion [4,5], friction [6], etc. In addition, the condensation process is critical for a wide range of industrial applications, such as refrigeration cycles [7], power plants [8], and water harvesting [9].

Based on macroscopic wettability (i.e., water contact angle), one would expect film-wise condensation on hydrophilic surfaces and drop-wise condensation on hydrophobic surfaces. However, the condensation behavior on the microscopic scale does not necessarily correlate with the expectation from the macroscopic measurement. For example, recent studies have revealed that water molecules form nanodroplets on surfaces that are defined as 'hydrophilic' by the macroscopic water contact angle measurements [1,2,10,11]. Therefore, the microscopic investigation of the water condensation process has attracted significant research interest.

Microscopic imaging of the water condensation process has been carried out by employing atomic force microscopy (AFM) and environmental scanning electron microscopy (ESEM). AFM is a valuable tool for imaging the size and distribution of condensed water

with nanometer resolution [2,12]. The ESEM is capable of characterizing growth dynamics of the micrometer-sized droplets with high temporal resolution [13–17]. A higher spatial resolution has been achieved by performing the transmission mode in ESEM, and thus the nucleation dynamics of water at the nanoscale level can now be observed [18]. However, the low temporal resolution of conventional AFM hinders its application in dynamic condensation studies, and the small imaging area resolved by the transmission mode ESEM makes it suitable only for characterizing nucleation or growth kinetics of individual water droplets. Thus far, the kinetics of the initial water condensation process on the global scale is still poorly understood.

Molecular dynamics (MD) simulations provide insights into the kinetics of water condensation on the nanoscale. Niu et al. reported that the film-wise mode possesses better heat transfer efficiency than the drop-wise mode at the nanoscale, indicating that the interfacial effect is dominant [19]. To facilitate computational efficiency, the high-density vapor medium was often used for simulation (i.e., saturated vapor medium at the temperature of 450 K or above). Yang et al. performed a water condensation simulation at 350~450 K. The results also exhibited a better heat transfer efficiency of film-wise condensation.

Moreover, the slower kinetics of water condensation onto the hydrophobic surface at a lower vapor temperature was observed [20]. Thus far, most previous simulations tuned the surface wettability by the water–surface interaction energy. Ranathunga et al. performed the MD simulations of water condensation on the self-assembled monolayers (SAMs) with different terminal groups. Aside from the water–SAM interaction energy, their results showed that the physicochemical properties of terminal groups play an essential role in the water condensation process [21]. These results verified the necessity to experimentally investigate functional groups' impact on initial water condensation.

Quartz crystal microbalance with energy dissipation (QCM-D) is a technology that dynamically detects mass changes with high sensitivity (less than 10 ng/cm$^2$) and provides viscoelastic information about the adsorbed mass [22]. Su et al. employed the QCM to record the kinetics of water condensation on nanostructured surfaces and characterized different condensation regimes [23]. Although only structural effects were discussed, their work demonstrated the great potential of QCM in studying the water–surface interaction at the initial condensation stage.

The purpose of this study is to explore the impact of the physicochemical properties of surfaces on the initial water condensation process. SAMs with different functional groups were utilized as model systems. We equipped the QCM-D with a precise dew point generator to allow the real-time characterization of the water condensation process. We attempted to evaluate the kinetics of water condensation and the evolution of the water–SAM interaction. Moreover, we also studied how the viscoelasticity of the growing water layer/nano droplet varies with head groups.

## 2. Materials and Methods

### 2.1. Preparation of Self-Assembled Monolayers (SAMs) on QCM-D Sensors and Water Contact Angle Measurements

The Au-coated sensors (a primary resonance frequency of 5 MHz) were first cleaned by a UV/ozone treatment for 20 min. Then, the sensors were immersed in ethanol for 30 min to remove the gold oxides formed during the UV/ozone treatment. The self-assembled monolayers (SAMs) of thiol derivatives were fabricated according to the previous paper [24]. Briefly, SAMs were fabricated by immersing the sensors overnight in ethanolic solutions containing the corresponding thiols at a concentration of 1 mM. After the immersion, the samples were carefully rinsed with pure ethanol to remove physisorbed thiol molecules from the gold sensor surface. After this, the sensors were dried with pure nitrogen. In this study, hydrophobic (C8 and CF$_3$), hydrophilic protein-adsorbing (OH), and hydrophilic protein-resistant [sulfobetaine, SB, and oligo(ethylene glycol), EG3-OH] SAMs were employed [25]. The chemical structures of thiols are shown in Table 1. All thiol molecules were purchased from Sigma-Aldrich (St. Louis, MO, USA) and used without further purification.

**Table 1.** Self-assembled monolayers used in this work and corresponding water contact angles.

| Abbreviation | Chemical Structure | Static Water Contact Angle (°) |
|:---:|:---:|:---:|
| C8 | HS-$(CH_2)_7$-$CH_3$ | 109 (2.3) [1] |
| CF3 | HS-$(CH_2)_2$-$(CF_2)_7CF_3$ | 114 (3.5) |
| OH | HS–$(CH_2)_{11}$–OH | 17 (0.9) |
| EG3-OH | HS–$(CH_2)_{11}$–(O–$CH_2$–$CH_2)_3$–OH | 37 (1.7) |
| SB | HS-$(CH_2)_{11}$-$N^+(CH_3)_2$-$(CH_2)_3$-$SO_3{}^-$ | 19 (1.4) |

[1] Number in parenthesis are standard deviations (*n* = 3).

Static water contact angle measurements were carried out using the contact angle analyzer (Phoenix Smart (A), SEO, Korea). A droplet of distilled water (3 µL) was placed on the surface of each sample at room temperature. The tangent and theta/2 methods were used to evaluate static water contact angles from the microscope images (Table 1).

### 2.2. QCM-D Measurement

In this work, a commercial QCM-D (QSense AnalyzerSystem, Biolin Scientific AB, Gothenburg, Sweden) was equipped with a precise dew point generator to supply humid $N_2$ gas with a dew point temperature of 35 °C (Figure 1). Before each water sorption experiment, dry $N_2$ gas was first flowed into the chambers for 1 h to remove the remaining water vapor. The humid $N_2$ gas was then introduced into the sample chamber via connection tubes heated at 50 °C to prevent the water condensation inside the tubes. Since the temperature of the sample chambers was held at 28 °C, the water vapor from the relatively hot and humid $N_2$ gas started to condense onto the sensors. Consequently, the water condensation process was dynamically monitored by the QCM-D.

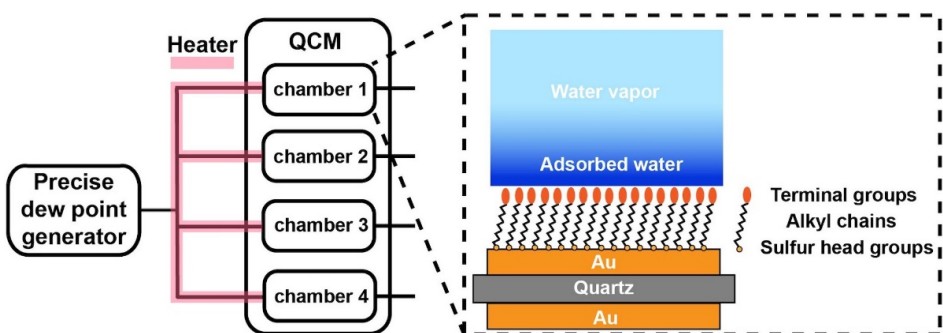

**Figure 1.** Schematic illustration of the QCM-D measurement in supersaturated condition.

### 2.3. QCM-D Data Analysis

2.3.1. Determination of the Mass and Thickness of Adsorbed Water on SAMs

When the added layer is thin, rigid, and homogeneously distributed over the sensor, the frequency change ($\Delta f$) can be converted to mass change ($\Delta m$) by the Sauerbrey relation Equation (1):

$$\Delta m = -C \cdot \Delta f / n, \tag{1}$$

where *n* is the overtone number and *C* is a mass sensitivity constant, which is approximately 17.7 ng/($cm^2 \cdot$Hz) for the sensors used in this work at room temperature.

In the case of viscoelastic films, the Sauerbrey equation underestimates the mass, and thus, viscoelastic modeling should be used. In the QCM-D technique, the dissipation factor *D* is also monitored, which is defined in Equation (2):

$$D = E_{diss} / 2\pi E_{strd}, \tag{2}$$

where $E_{diss}$ and $E_{strd}$ are the energy dissipated during a single oscillation period and the energy stored in the oscillator, respectively.

Compared to rigid films, more energy is dissipated from the system when viscoelastic films are in contact with the surface, resulting in higher dissipation shifts. Therefore, the change in dissipation ($\Delta D$) measures the rigidity of the added materials onto the sensor. The viscoelastic layer can be distinguished from the rigid layer when the $\Delta f$ and $\Delta D$ of the different overtones diverge. This characteristic was exhibited in Figure S1, indicating the viscoelasticity of the adsorbed water layer. Therefore, the model for single viscoelastic films in the air was used in this work to investigate the adsorbed water layers on different SAM surfaces, where the frequency change normalized per overtone $\Delta f/n$, is proportional to the square of the overtone order $n^2$ according to Equation (3):

$$\frac{\Delta f}{n} = \frac{-2f_0^2 m_f}{Z_q}\left[1 + \frac{1}{3}\frac{Z_q^2}{Z_f^2}\left(\frac{m_f}{m_q}n\pi\right)^2\right],\tag{3}$$

where $Z_q$, $Z_f$, $f_0$, $m_f$, and $m_q$ are the acoustic impedance of quartz ($8.8 \times 10^6$ kg m$^{-2}$ s$^{-1}$), the acoustic impedance of the film, the fundamental frequency ($\approx$5 MHz), the mass of the film per unit area, and the areal mass density of the crystal, respectively. For simplicity, Equation (3) can also be presented using Equation (4):

$$\Delta f_n = -am_f\left(1 + bm_f^2 n^2\right) = -am_f - abm_f^3 n^2,\tag{4}$$

where $a = 2f_0^2/Z_q$ and $b = Z_q^2\pi^2/\left(3Z_f^2 m_q^2\right)$.

The baseline of the resonant frequency is defined as the resonant frequency of the sensors coated with SAMs in dry N$_2$ gas. The frequency shift due to the adsorption of water onto the sensor ($\Delta f_w$) was calculated according to Equation (5):

$$\begin{aligned}\Delta f_w = f_{wet} - f_{dry}\\ = -a(m_{wet} - m_{dry}) - abn^2(m_{wet}^3 - m_{dry}^3)\\ = -am_w - abn^2(m_{wet}^3 - m_{dry}^3),\end{aligned}\tag{5}$$

where $f_{wet}$, $f_{dry}$, $m_{wet}$, $m_{dry}$, $m_w$ are the frequencies of the sensors with and without adsorbed water, the masses of the wet and dry SAM-functionalized sensors, and the mass of adsorbed water, which can be obtained by linear extrapolation to the intercept of the zeroth overtone, where $\Delta f_{w\ (n\to0)}/n = -am_w$, respectively. It is worth noting that the term $n^2$ was replaced by $n^\beta$, where $\beta$ was adjusted to give the best linear fit.

To facilitate the comparison of the water adsorption on each interface, the thickness $t$ was calculated by Equation (6) under the premise of two assumptions: (i) the adsorbed water has constant density, and (ii) the adsorbed water is homogenous on the surface and has a uniform thickness:

$$t = \frac{\Delta m}{\rho},\tag{6}$$

where $\Delta m$ is the mass of adsorbed water per unit area calculated by the viscoelastic model and $\rho$ is the density of water (0.9998 (g/cm$^3$)).

## 2.3.2. Analysis of Viscoelasticity of Adsorbed Water

The acoustic impedance of the film $Z_f$ in Equation (3) encompasses the storage and loss moduli, and by multiplying the numerator and denominator with the complex conjugate of $G_f$, Equation (7) is obtained as shown:

$$\frac{1}{Z_f^2} = \frac{1}{\rho_f G_f} = \frac{\overline{G}}{\rho_f\left|G_f\right|^2} = \frac{1}{\rho_f}\frac{G_f' - iG_f''}{\left(G_f'\right)^2 + \left(G_f''\right)^2}\tag{7}$$

The rheological property of the adsorbed water film can be evaluated by the storage and loss moduli $G'/G''$ ratio [26], which is given by Equation (8) as follows:

$$\frac{G'}{G''} = -\frac{\Delta f + \frac{2f_0 n m_f}{Z_q}}{\Delta \Gamma} = -\frac{\Delta f + n\left(\frac{\Delta f}{n}\right)_{extrap}}{\Delta \Gamma},$$ (8)

where $(\Delta f/n)_{extrap}$ and $\Delta \Gamma$ are the normalized frequency change extrapolated to overtone zero and the change in bandwidth of the resonance peak, which can be expressed by dissipation $D$ according to $\Gamma = Df/2$, respectively.

## 3. Results and Discussion

### 3.1. Kinetics of the Water Condensation onto SAMs

Figure 2a shows noticeable differences in the kinetics for various SAMs during the water adsorption process. The hydrophobic SAMs (C8 and CF$_3$) showed smaller frequency shifts, whereas more significant frequency shifts were seen from the hydrophilic SAMs (OH, SB, and EG3-OH). The reason why water adsorbs more on hydrophilic surfaces in the same vapor medium has been revealed through theoretical studies. The wettability of surfaces can strongly affect the nucleation process [27] and heat transfer [19,20] during water adsorption. According to the classical nucleation theory, the hydrophobic surfaces have less probability of forming water clusters. The hydrophobic surfaces have less efficient heat transfer as compared to the hydrophilic surfaces in the thickness range of 5–20 nm [28]. Therefore, the condensation of water onto the hydrophobic surfaces has slow kinetics. Then, the dissipation shifts correspond to the changes in the viscoelastic properties of water adsorbed onto the SAM surfaces. As shown in Figure 2b, the dissipation shifts for each SAM increased at different rates during the water adsorption process. This indicates that a change from a rigid state to a more viscoelastic state has occurred as additional water layers form.

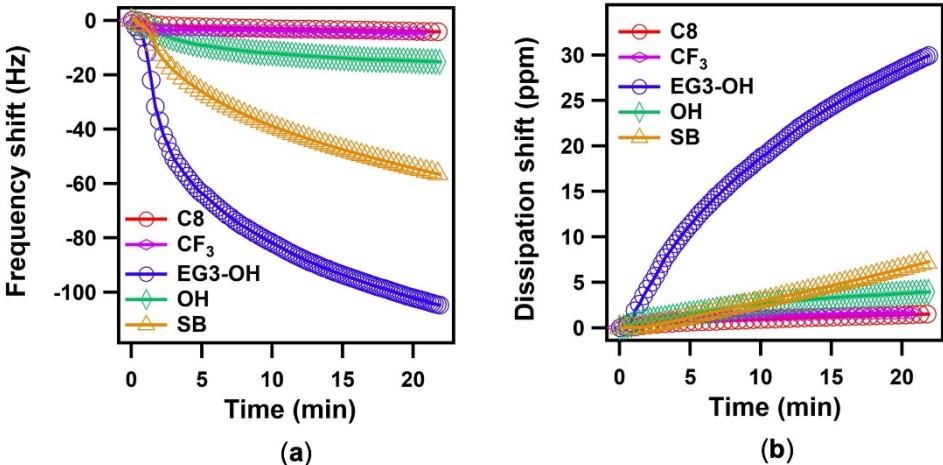

**Figure 2.** Changes in frequency (**a**), and dissipation (**b**) as a function of time [*n* (overtone) = 3] after the injection of water vapor.

The added layer observed in this work is viscoelastic, which is signified by an evident spread of the $\Delta f$ and $\Delta D$ of the different overtones (Figure S1). This will lead to an underestimation of mass by the Sauerbrey relation; therefore, a model for the single viscoelastic film in the air was used to correct the frequency change (Figure S2). For the convenience of comparison, the obtained mass was converted into the thickness of water by Equation (6), under the assumption that the density and distribution of the water layer are homogenous. Figure 3 shows the total thickness of the adsorbed water layers on different SAMs. The water adsorbed slowly on the hydrophobic SAMs and the thickness was saturated. On the other hand, water molecules kept condensing on the hydrophilic SAMs, which is consistent

with the results of the previous MD simulation [27]. In their results, water condenses slowly as nanoscale droplets onto a hydrophobic surface while the water layer develops quickly in film-wise mode on a hydrophilic surface. In addition, their results also suggested that the kinetics of water adsorption on hydrophilic surfaces was characterized by an initial rapid increase in layer thickness, followed by the gradual formation of the subsequent water layers. The initial fast adsorption can be attributed to the higher efficiency of interfacial heat transfer on hydrophilic surfaces [19,20]. As water condensation proceeds, the water film grows, and the condensate's bulk thermal resistance gradually becomes dominant over the interfacial thermal resistance, resulting in a slowdown of kinetics. It should be noted that there were some differences in the kinetics of adsorption among hydrophilic SAMs, which is likely due to the different hydration manner of SAM terminal groups. This issue will be discussed later.

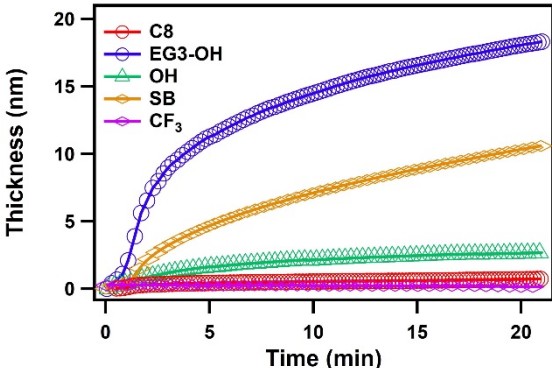

**Figure 3.** The thickness of the water layers on C8, CF$_3$, EG3-OH, OH, and SB SAMs plotted as a function of time.

### 3.2. Interactions between Adsorbed Water and SAMs

The information on the water–SAM interactions and viscoelastic properties of adsorbed water can be obtained by comparing the dissipation change ($\Delta D$) versus the frequency change ($\Delta f$). A steep $\Delta D/\Delta f$ curve indicates the softness of the water layer and the weak interaction between water and the SAMs. As seen in Figure 4a, all the $\Delta D/\Delta f$ curves consisted of two parts, which are an initial curving slope and a following linear slope indicating two distinct states of water (interfacial and bulk water)

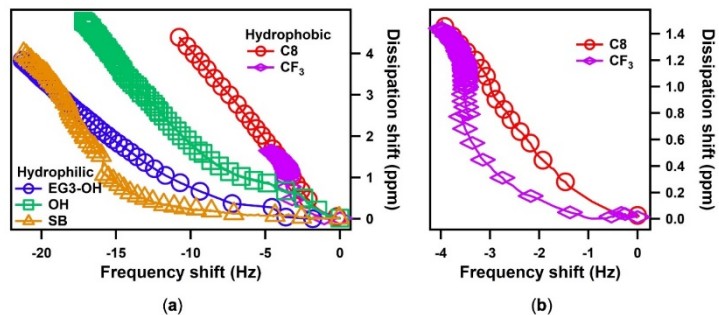

**Figure 4.** (**a**) The energy dissipation shift, $\Delta D$, plotted as a function of the frequency shift, $\Delta f$, for the different SAMs; (**b**) The enlarged figure of $\Delta D/\Delta f$ plots of hydrophobic SAMs.

The hydrophobic SAM shows steep curves at the initial stage compared with the hydrophilic SAM, indicating the weak interactions between the hydrophobic surfaces and water [21]. In contrast, the hydrophilic SAM exhibited a milder curve resulting from a strong interaction between water and the surface. Among the hydrophilic SAMs, the protein-resistant SAMs (SB and EG3-OH) showed even milder curves at the initial stages than protein-adsorbing OH SAM, implying thicker and stiffer water layers formed on SB and EG3-OH SAMs. These results are in good agreement with previous findings suggesting that the interfacial water

on SB and EG3-OH acts as a physical barrier in preventing protein adsorption [24,29–36]. It should be noted that the curve for SB SAM had an even lower slope than that of EG3-OH SAM at the first stage. This finding supports the simulation results that water binds more strongly with the sulfobetaine than the oligo(ethylene glycol) (OEG) group [37,38]. It is also noteworthy that the macroscopic contact angles decreased in the order of EG3-OH SAM > SB SAM > OH SAM. However, the strengths of water–SAM interactions decreased in the order of SB SAM > EG3-OH SAM > OH SAM, which is consistent with the results of the molecular simulations [21]. This finding indicates that the macroscopic water contact angle cannot predict the local structure of water at SAM–water interfaces.

On the other hand, at the first stage curves for hydrophobic surfaces, $CF_3$ SAM showed a more moderate curve than the C8 SAM (Figure 4b), indicating stronger water–$CF_3$ SAM interaction. This seems perplexing because $CF_3$ SAM has a larger water contact angle than C8 SAM. However, this finding is consistent with the hydration manner on a microscopic scale. Previous theoretical results revealed that the C-F bond has a stronger binding with individual water molecules than the C-H bond because of the stronger dipole moment of the $CF_2$ and $CF_3$ groups. However, the larger molecular volume and sparse packing density of the $CF_3$ unit lead to the higher atomic roughness of the $CF_3$ SAM surface, resulting in a higher energetic cost of cavity formation in water. This energetic penalty offsets the C-F bond's greater affinity for water [21,39,40]. Subsequently, when water accumulates more on the surface, water molecules with strong tendencies to maintain their number of hydrogen bonds are likely to interact more with themselves, resulting in the greater hydrophobicity of $CF_3$ SAM on a macroscopic scale.

### 3.3. The Viscoelasticity of Adsorbed Water on SAMs

To further investigate the viscoelastic properties of adsorbed water, the ratio of the storage and loss moduli $G'/G''$ was calculated according to Equation (8). For a rigid film, a high $G'/G''$ value is expected due to the dominance of the elastic behavior. In contrast, a viscoelastic film gives a low $G'/G''$ value close to zero due to the dominant viscous-like property. As shown in Figure 5, the $G'/G''$ ratios for all SAMs had high values at film thicknesses of less than 1 nm and then dropped to around zero as the thickness increased. This drastic decrease indicates the change of viscoelastic properties of the adsorbed water from interfacial to bulk states. For hydrophilic protein-adsorbing OH SAM and hydrophobic C8 and $CF_3$ SAMs, the ratios of $G'/G''$ dropped quickly with increasing the film thickness and approached around zero at a thickness of less than 1 nm. On the other hand, $G'/G''$ for the EG3-OH and SB SAM also decreased quickly but maintained specific values in a less than 2 nm range and then dropped to around zero at around 3 nm. These results suggest that SAMs with protein-resistant properties have a stiffer interfacial water layer with a thickness of 2–3 nm.

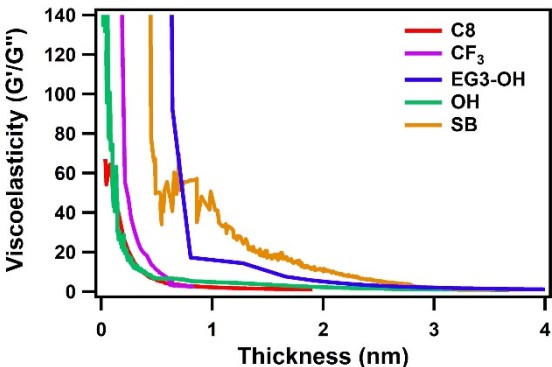

**Figure 5.** Ratio $G'/G''$ of the Au sensors coated with SAM as a function of the thickness of adsorbed water.

The difference in the viscoelasticity of interfacial water originates in the difference in the water–SAM interactions. Combining our previous findings and the previously reported results, we show a schematic illustration of water interacting with different SAMs in Figure 6. The tightly-bound water was discovered on both OEG and SB SAMs by vibrational spectroscopy [41]. In the case of SB SAM, water molecules strongly interact with positively and negatively charged groups via electrostatic interactions [37]. Such strong water–SAM interaction can modulate the properties of the subsequent water molecules in the interfacial region. As for the EG3-OH SAM, water molecules penetrate the terminal groups and interact strongly with the oxygen atoms in the OEG chains via hydrogen bonding. The mixture of OEG chains and trapped water acts as a template to modulate the water molecules in the interfacial region [32]. Hayashi et al. observed water-induced repulsive force in the range of 4–6 nm between the probe and substrate pairs functionalized with EG3-OH SAM, SB SAMs, and other protein-resistant SAMs [24,30,32,34,36,42,43]. They stated that the repulsive force originated from the different mechanical properties of interfacial water. Their model is substantiated by the observed highly viscous interfacial water layer on EG3-OH and SB SAMs.

**Figure 6.** Summary of this work including the results of the water contact angle measurements, illustrations of microscopic views of water condensation, the film growth manners, water–molecule interactions, and the hydration states of the SAMs.

In contrast, OH SAM induces the formation of a less viscous interfacial water layer in a shorter range, which is consistent with the weak water–SAM interaction obtained by simulations [29,44] and the absence of water-induced force measured by force measurement [24]. Due to the weak water- SAM interaction, the adsorbed interfacial water droplet exhibited relatively low viscosity for hydrophobic SAMs. Our results demonstrated that the strong water–SAM interaction modulates the viscoelasticity of water molecules in the interfacial region, extending to 2–3 nm. In contrast, the weak interaction has a more negligible effect on the viscoelasticity of water adjacent to the surfaces.

## 4. Conclusions

By employing the QCM-D, we performed real-time measurements of water condensation on self-assembled monolayers (SAMs) with thiol molecules with different functional groups. The kinetics of the water condensation exhibited critical dependence on the physicochemical properties of the surface. Water continuously adsorbs onto the hydrophilic SAMs, whereas the condensations are saturated at specific points for the hydrophobic SAMs. These findings support the simulation results that water condensation exhibits different regimes at the nanoscale level depending on the surface–water interactions.

Comparison of the $\Delta D / \Delta f$ values revealed a rigid water film formed on the hydrophilic SAMs and loosely bound water (probably nanodroplet) on the hydrophobic surfaces, in agreement with the condensation processes evaluated by the simulation. It was also found that the strengths of water–SAM interactions do not agree with the behaviors expected from macroscopic contact angles but rather agree with the predictions from molecular simulations. Our results indicated that the macroscopic water contact angle is not always a good indicator for the nanoscale condensation process.

From the viscoelastic analyses, we observed the presence of a stiffer interfacial water layer in the range of 2–3 nm on protein-resistant SAMs. In contrast, the thicknesses of the interfacial water on either hydrophilic protein-adsorbing OH SAM or hydrophobic SAMs were small (<1 nm). By combining our observations and results from the literature, we concluded that the water–SAM interaction is a primary factor in modulating the viscoelasticity of the interfacial water layer, and the range of modulation also depends on the terminal groups. This work sheds new light on understanding water condensation behaviors at the nanoscale. Thus, we expected that our findings would provide new insights into the design of the surfaces of anti-fogging glasses, self-cleaning surfaces, microfluidic systems, etc.

**Supplementary Materials:** The following supporting information can be downloaded at: https://www.mdpi.com/article/10.3390/micro2030033/s1, Figure S1. Changes in frequency and dissipation (n = 3, 5, 7) as a function of time during water adsorption onto C8 SAM; Figure S2. The mass of adsorbed water is determined by the extrapolation of Δf/n to the zeroth overtone.

**Author Contributions:** Conceptualization, T.H. and E.A.Q.M.; methodology, S.S., R.C. and T.H.; software, S.S., G.V.L., R.C., E.A.Q.M. and T.H.; writing—original draft preparation, S.S. and T.H.; writing—review and editing, G.V.L., E.A.Q.M. and T.H.; project administration, T.H. All authors have read and agreed to the published version of the manuscript.

**Funding:** This work was supported by the JSPS KAKENHI grant (Grant Number JP22H04530, JP21H05511, JP20H05210, and JP19H02565). This work was performed under the "Five-star Alliance" Research Program in "NJRC Mater. & Dev.".

**Institutional Review Board Statement:** Not applicable.

**Informed Consent Statement:** Not applicable.

**Data Availability Statement:** Our research activities are summarized in http://lab.spm.jp/ accessed on 23 June 2022.

**Acknowledgments:** We thank Kazue Taki for the administration of this project.

**Conflicts of Interest:** The authors declare no conflict of interest.

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
