# Peer review of "Experimental Characterization of Water Condensation Processes on Self-Assembled Monolayers Using a Quartz Crystal Microbalance with Energy Dissipation Monitoring"

_2673-8023, doi:10.3390/micro2030033_

Round 1
Reviewer 1 Report
This paper reported water condensation processes on different self-assembler monolayers. The processes were monitored using QCM-D. Performed experiments and their explanation look interesting. Overall, the work was well conducted, and the results were adequately presented and discussed. The manuscript can be considered for publication after minor corrections.
Comments:
11) Was the same preparation procedure used before QCM-D and contact angle measurements? If samples were not dried into the chambers by N2 gas before contact angle measurements, how could this affect the accuracy of the measurements?
22) The results of contact angle measurements should be provided with errors.
Author Response
1) Was the same preparation procedure used before QCM-D and contact angle measurements? If samples were not dried into the chambers by N2 gas before contact angle measurements, how could this affect the accuracy of the measurements?
Author reply:We thank you for your comment. We performed the WCA measurements with the QCM sensors, meaning that the samples for QCM-D and WCA measurements were identical. In the final step of the sample preparation, we dried the samples with pure N2. We added the description in the main text.
After this, the sensors were dried with pure nitrogen.
2) The results of contact angle measurements should be provided with errors.
Author reply:We appreciate this comment. The errors (standard deviation) has been indicated in Table 1. The values of the contact angles in Fig. 6 were removed to avoid the confusion.
We thank you again for your suggestion.
Reviewer 2 Report
Authors in this study have explored a real-time characterization of water condensation on self-assembled monolayers (SAMs) with different functional groups using quartz crystal microbalance with energy dissipation monitoring (QCM-D). Studies such as the kinetics of water condensation, contact angle, and viscoelasticity measurements are carried out to establish the mechanism. The work is interesting and the selection of functional groups for the study is also appropriate. Therefore I recommend the acceptance of the manuscript with the following minor revision.
1. Figure 5. Ratio of G'/G'' for sulfobetaine is found to be different "On the other hand, G′/G′′ for the protein-resistant SAMs 276 (SB and EG3-OH) also dropped quickly but maintained specific values in less than 2 nm 277 range and then dropped to around zero at around 3 nm." This statement of the authors is not accurately reflected in the graphs of SB. Justify
2. Figure 6 caption is not accurate. Also the description and presentation of the schematic figure is confusing. The clear purpose of the figure must be reflected.
3. The outcome of the study in the conclusion section must be rewritten.
Author Response
- Figure 5. Ratio of G'/G'' for sulfobetaine is found to be different "On the other hand, G′/G′′ for the protein-resistant SAMs 276 (SB and EG3-OH) also dropped quickly but maintained specific values in less than 2 nm 277 range and then dropped to around zero at around 3 nm." This statement of the authors is not accurately reflected in the graphs of SB. Justify
Author reply: We thank you for your comments. Actually, G'/G'' for EG3-OH and SB show similar trend. We modified the main text as the following:
On the other hand, G′/G′′ for the EG3-OH and SB SAM also dropped quickly but maintained specific values in less than 2 nm range and then dropped to around zero at around 3 nm.
- Figure 6 caption is not accurate. Also the description and presentation of the schematic figure is confusing. The clear purpose of the figure must be reflected.
Author reply: We appreciate your comment. The figure was revised. We also modified the caption as the following.
Summary of this work including the results of the water contact angle measurements, illustrations of microscopic views of water condensation, the film growth manners, water-molecule interactions, and the hydration states of the SAMs.
- The outcome of the study in the conclusion section must be rewritten.
Author reply: Thank you for your comment. According to your advice, we modified the outcomes of this work as the following:
This work shed new light on understanding water condensation behaviors at the nanoscale. Thus we expected that our findings would provide new insights into the design of the surfaces of anti-fogging glasses, self-cleaning surfaces, microfluidic systems, etc.
We thank you again for your helpful suggestion.